# Identification of Host PDZ-Based Interactions with the SARS-CoV-2 E Protein in Human Monocytes

**DOI:** 10.3390/ijms241612793

**Published:** 2023-08-14

**Authors:** Antonia Ávila-Flores, Juan José Sánchez-Cabezón, Ane Ochoa-Echeverría, Ana I. Checa, Jorge Rosas-García, Mariana Téllez-Araiza, Sara Casado, Rosa Liébana, Teresa Santos-Mendoza, Isabel Mérida

**Affiliations:** 1Department of Immunology and Oncology, Spanish National Centre for Biotechnology, 28049 Madrid, Spain; jj.sanchez@cnb.csic.es (J.J.S.-C.); ane.ochoa@cnb.csic.es (A.O.-E.); acheca@cnb.csic.es (A.I.C.); scasado@cnb.csic.es (S.C.); rliebana@cnb.csic.es (R.L.); 2Laboratory of Transcriptomics and Molecular Immunology, Instituto Nacional de Enfermedades Respiratorias Ismael Cosío Villegas, Mexico City 14080, Mexico; jorge.rosas@cinvestav.mx (J.R.-G.); mariannt87@hotmail.com (M.T.-A.); tsantos@iner.gob.mx (T.S.-M.)

**Keywords:** SARS-CoV-2, envelope (E) protein, PDZ, monocyte, innate immune response, syntenin, ZO-2, IL-16

## Abstract

Proteins containing PDZ (post-synaptic density, PSD-95/disc large, Dlg/zonula occludens, ZO-1) domains assemble signaling complexes that orchestrate cell responses. Viral pathogens target host PDZ proteins by coding proteins containing a PDZ-binding motif (PBM). The presence of a PBM in the SARS-CoV-2 E protein contributes to the virus’s pathogenicity. SARS-CoV-2 infects epithelia, but also cells from the innate immune response, including monocytes and alveolar macrophages. This process is critical for alterations of the immune response that are related to the deaths caused by SARS-CoV-2. Identification of E-protein targets in immune cells might offer clues to understanding how SARS-CoV-2 alters the immune response. We analyzed the interactome of the SARS-CoV-2 E protein in human monocytes. The E protein was expressed fused to a GFP tag at the amino terminal in THP-1 monocytes, and associated proteins were identified using a proteomic approach. The E-protein interactome provided 372 partners; only 8 of these harbored PDZ domains, including the cell polarity protein ZO-2, the chemoattractant IL-16, and syntenin. We addressed the expression and localization of the identified PDZ proteins along the differentiation of primary and THP-1 monocytes towards macrophages and dendritic cells. Our data highlight the importance of identifying the functions of PDZ proteins in the maintenance of immune fitness and the viral alteration of inflammatory response.

## 1. Introduction

The PDZ (post-synaptic density, PSD-95/disc large, Dlg/zonula occludens, ZO-1) domain is a small (80–110 amino acids (aa)) modular region present in around 150 distinct human proteins [1]. PDZ domains facilitate the assembly of multiprotein complexes through the recognition of consensus PDZ-binding motifs (PBMs) present at the C-terminal regions of their partners [2]. PDZ-containing proteins participate in the organization of specialized subcellular compartments that facilitate cell polarity and cell–cell communication. Unsurprisingly, alterations in PDZ networks are frequently related to pathological states [3]. Defects in PDZ-containing proteins affect epithelial cell polarity, favoring epithelial–mesenchymal transition [4]; remarkably, 124 PDZ domains are linked to cancer [3]. Similarly, neuronal PDZ network alterations induce neurodegenerative and mental disorders [5]. PDZ-regulated functions are less studied in immune cells, but several polarity proteins are shared in epithelial and immune cells [6,7]. Defects in immune responses, including hyperproliferation and memory imbalance of T-cell subsets or limited production of reactive oxygen species in macrophages, are associated with defects in PDZ-based interactions [8,9,10].

Targeting of PDZ proteins by viral pathogens represents a potent strategy to hijack the host machinery during infection to facilitate viral replication and expansion [11]. The impact of viral proteins on host cellular functions is, in many cases, directly related to their ability to target PDZ proteins. For instance, only the high-risk, oncogenic human papilloma virus (HPV) strains interact with PDZ proteins, causing loss of epithelial cell polarity. PBMs are present in the Tax proteins of human T-lymphotropic (HTLV) type 1 viruses that cause adult T-cell leukemia/lymphoma (ATLL), but they are absent in type 2 viruses that do not cause lymphoproliferative diseases [2]. Interestingly, alterations caused by the disruption of PDZ-based interactomes may appear in the long term after viral remission, like those causing cancer through the HPV E6 protein [12]. The presence of PBMs in zoonotic viruses is a clear determinant of their higher infectivity when compared with similar human strains. Thus, the NS1 protein of avian influenza A shows high reactivity with PDZ domains compared with the less pathogenic human strains [13]. β-Coronaviruses that have caused zoonotic epidemic infections, including the currently circulating severe acute respiratory syndrome coronavirus 2 (SARS-CoV-2), harbor an envelope (E) protein that contains a PBM [14]. Studies on SARS-CoV-1 have shown that the E-protein PBM is indeed a pathogenicity factor since a virus modified to lack this domain was not capable of inducing disease in animal models [15].

SARS-CoV-2 infection has caused the COVID-19 pandemic responsible for over 6.5 million deaths worldwide. COVID-19-related deaths and the long-term secondary effects linked to this infection mainly result from a dysregulated release of pro-inflammatory cytokines, which cause severe damage to organs [16]. SARS-CoV-2 infects epithelia, but many reports suggest that it can also infect cells from the innate immune response, including monocytes, dendritic cells, and alveolar macrophages [17,18]. SARS-CoV-2 monocyte infection, albeit sterile, leads to inflammasome formation, followed by pyroptosis and cell death [19]. Early studies with the SARS-CoV-1 E protein characterized its capacity to form protein–lipid pores in the membrane to allow ion transport, an activity linked to the activation of the NLRP3 inflammasome [20]. However, the ability of the SARS-CoV-2 E protein to hijack PDZ-containing targets in monocytes is so far unknown. Their identification might offer valuable clues to understand how SARS-CoV-2 alters inflammatory responses, and to delineate new therapeutic strategies. Here, we expressed the SARS-CoV-2 E protein as a GFP-fused recombinant protein in a human monocytic cell line and investigated the PDZ-dependent SARS-CoV-2 E-protein interactome. We found that the viral protein pulls down distinctive proteins, some of which have never been characterized in immune cells. Our findings provide new hints linking viral PDZ targeting with inflammatory response and highlight the importance of PDZ proteins in immune fitness.

## 2. Results

### 2.1. Construction and Expression of a Recombinant SARS-CoV2 E-Protein GFP Tagged at the Amino Terminal Region

The SARS-CoV-2 envelope (E) protein is the smallest (75 aa) structural protein of the virus. It contains a hydrophilic N-terminal domain (NTD), followed by a hydrophobic transmembrane domain (TMD) and a long hydrophilic C-terminal domain (CTD) [21] (Figure 1A). The SARS-CoV-2 E-protein sequence is highly similar to that of the SARS-CoV-1, including the presence of a -DLLV C-terminal sequence that constitutes a type II PBM [22] (Figure 1A). We used the cloned ORF of the SARS-CoV-2 E protein [22] to generate a recombinant protein with a GFP tag at the amino terminal. The pDONR207 SARS-CoV-2 E plasmid [23] was recombined with the pEZYeGFP vector [24] to produce the pEZYeGFP-SARS-CoV-2 E plasmid, which encodes a GFP recombinant protein with a predicted molecular size of 37.11 kDa. The pEZYeGFP-SARS-CoV-2 E plasmid was transfected into the human monocytic cell line THP-1, as well as into HEK-293T and Caco-2 cells. The expression of the protein analyzed by immunoblot revealed a double band near 37 kDa (Figure 1B and Appendix A). The use of different protease inhibitors during extraction procedures argues against the possibility that the minor size protein corresponded to a degradation product. Probably, the slow migrating band corresponds to a post-translationally modified version of the protein, similar to that reported for the SARS-CoV-1 E protein. The presence of hydrophobic residues in the transmembrane region, as well as palmitoylation on cysteine residues, in the SARS-CoV-1 E protein are known to produce different migrating bands when analyzed by gel electrophoresis [25]. Noteworthy, these motifs are conserved in the SARS-CoV-2 E protein (Figure 1A). 

We next investigated the subcellular distribution of the GFP-SARS-CoV-2 E protein in THP-1, HEK-293T, and Caco-2 cells by confocal microscopy. The E protein had a punctuate distribution pattern; it accumulated around the nuclear region and partially, at the cell membrane. Although similar, the pattern was not identical in the 3 cell lines (Figure 1C and Appendix A). Nonetheless, the accumulation near the nuclear region was similar in all the cases and suggested Golgi localization. Accordingly, it was reported that a recombinant SARS-CoV-1 E protein expressed in HeLa cells localizes to the Golgi region [26]. We next examined the distribution of the SARS-CoV-2 E protein using the Golgi marker giantin in THP-1 and HEK-293T cells (Figure 1D). Analysis of the images indicated co-localization of E protein with giantin (Figure 1E–G). Therefore, we conclude that a pool of the GFP-SARS-CoV-2 E protein is localized in the Golgi region. 

### 2.2. Expression of the GFP-SARS-CoV-2 E Protein Affects the Viability of THP-1 Cells

We observed that THP-1 cells were highly sensitive to the expression of the recombinant GFP-E protein. THP-1 cell sensitivity was corroborated using flow cytometry to evaluate cell viability in the cells positive for GFP expression. Compared to cells expressing GFP, those expressing the GFP-SARS-CoV-2 E protein (GFP-E), showed an increase in death after 6 h of being transfected (Figure 2A,B). The negative effects of the GFP-E protein over cell viability agree with the reported functions of the SARS-CoV-1 E protein as a viroporin and virulence determinant [14,15].

The formation of inflammasomes is a characteristic feature of myeloid cells when they sense infection [27]. Thus, we determined whether the expression of the SARS-CoV-2 E protein promotes cleavage/degradation of the Gasdermin D protein (GSDMD), a step that initiates the formation of the NRLP3 inflammasome. Western blot analysis showed increased degradation of GSDMD in the cells transfected with the pEZYeGFP-SARS-CoV-2 E plasmid, compared to those transfected with the pEZYeGFP control plasmid (Figure 2C, left panel). The pEZYeGFP-SARS-CoV-2 E plasmid-transfected cells were sorted to separate the culture into GFP positive and GFP negative cells; only the GFP positive cells (expressing the GFP-SARS-CoV-2 E protein), displayed increased degradation of GSDMD (Figure 2C, right panel). These data indicated that in monocytes, the SARS-CoV-2 E protein triggers the formation of inflammasomes.

### 2.3. Proteomic Analysis of the GFP-SARS-CoV-2 E Protein in THP-1 Cells 

Experiments using two hybrid assays, as well as a screening of PDZ domains containing libraries with peptides of the C-terminal region of the E protein, have identified a wide variety of PDZ-containing proteins that can potentially be targeted by the SARS-CoV-2 E protein. A better understanding of the SARS-CoV-2 E protein–host interactome might provide clues to know COVID-19 disease mechanistically. This is particularly relevant in monocytes, since these cells play an important role in the immune response against SARS-CoV-2. The GFP-SARS-CoV-2 E protein was expressed in THP-1 monocytes, purified using GFP-trap beads, and the associated proteins were identified by liquid chromatography coupled to Triple TOF Mass Spectrometry. Cells expressing the GFP vector were used as controls, and only those with a *q*-value <0.01 were selected This strategy enabled the discrimination of protein contaminants from true protein binders. Peptides corresponding to 372 proteins were identified only in the GFP-E-protein pull-down. We explored the potential biological relevance of the identified ligands of the E protein by STRING (Search Tool for the Retrieval of Interacting Genes/Proteins) analysis [28]. The significantly enriched GoTERM (Gene Ontology Term Enrichment) categories of biological process and cellular distribution, including the number of proteins in each one, are highlighted in Figure 3. Proteins are listed in the additional analysis included in Appendix A.

The highest enrichment scores for the biological processes category were related to immune response-related functions, including leukocyte degranulation and activation of myeloid cells. Some of the identified proteins included negative regulators of the Toll-like receptors (TLRs), like the Toll-interacting protein TOLLIP; adhesion molecules like integrin alpha L and CD44; or tyrosine kinases like Syk, Lyn and HCK (Figure 3A, Appendix A). It is noteworthy that 53 interactors were functionally linked to viral processes, including ribosomal components and RNA-binding proteins (MOV10A, RAE1, and STAU1). Further, 47 interactors were associated with exocytosis, including members of the exocyst complex and syntaxins (Appendix A). Near 8% of the associated proteins were functionally related to membrane-located processes, including membrane targeting and translation (Figure 3A and Appendix A). In agreement, cellular compartment enrichment identified vesicle and secretory pathways, as well as ribosomes and endomembrane networks (Figure 3B; Appendix A). The biological process and subcellular distribution of the partners coincide with their categorization based on the reactome pathway and tissue expression (Figure 3C,D). Remarkably, nearly 100 partners of the SARS-CoV-2 E protein were related to specialized cell–cell communication structures, including cell junctions and synapses (Figure 3B; Appendix A). Further analysis of this cluster revealed enrichment in ribosome components, as well as proteins from the adherent junctions. These last included the PDZ proteins scribble homolog (Scrib), zonula occludens/tight junction protein-2 (ZO-2/TJP-2), and protein lin-7 homolog A (Lin-7A), as well as PDZ interactors, like junctional adhesion molecule A (F11R), catenin delta (CTNND1), and junctional protein associated with coronary artery disease (KIAA1462) (Appendix A). The abundance of proteins related to cell–cell communication correlates with their categorization as brain-expression-specific (Figure 3D).

### 2.4. Identification of the PDZ Partners of the GFP-SARS-CoV-2 E Protein in Monocytes

In order to focus only on the SARS-CoV-2 E protein PDZ-dependent interactome, the proteomic results were next compared with 155 human proteins containing PDZ motifs [29]. In addition to Scrib, ZO-2 and Lin-7A, the new analysis identified other 5 PDZ proteins: syntenin, MPP1 (membrane palmitoylated protein 1), pro-IL-16 (pro-interleukin 16), SIPA1L1 (signal-induced proliferation-associated 1-like protein 1), and the post-synaptic scaffolding protein delphilin (Table 1). 

Some of the 8 identified PDZ partners of protein E have been previously reported as targets of other viral proteins, and in some cases, this interaction has been related to viral pathogenesis (Table 1). Remarkably, syntenin was formerly identified as a partner of the SARS-CoV-1 E protein [15], which validates our experimental findings. Another interactor, MMP1 (also known as 55 kDa erythrocyte membrane protein; EM55), belongs as MPP1 to the subfamily of MPPs (membrane palmitoylated proteins), a group enclosed into the large family of scaffold and multidomain proteins known as MAGUKs (membrane-associated guanylate kinases) [30]. PALS1 (protein associated with lin-seven-1), which was identified early as a SARS-CoV-1 E protein partner in epithelial cells [31], is a different member of the MPP subfamily. PALS1 was not present in our proteomic assays, but we identified its interactor, the polarity protein Lin-7A, which in epithelial cells distributes the receptor tyrosine kinase Let-23 to the basolateral membrane [32]. The expression of Lin-7A in the innate immune system has not been reported.

Table 1 Only 8 out of the 372 identified proteins harbor PDZ domains. The table summarizes the main function attributed to these proteins, the number of PDZ domains and additional domains present. Reported interactions with other viral proteins are also indicated. 

Our identification of Scrib as a potential partner of the SARS-CoV-2 E protein adds this protein to the large list of viral proteins that interact with this adaptor molecule that facilitates key molecular interactions at distinctive subcellular regions [33]. Scrib belongs to the LAP (LRR and PDZ) family; it contains 16 Leucine Rich Repeats (LRR), two LAP-specific regions and four PDZ domains. In epithelial cells, Scrib forms a complex that localizes to the basolateral membrane which is essential for the establishment of cell polarity [34], whereas synaptic terminals regulate vesicle clustering and release [35]. Scrib also participates in the organization of the immunological synapse in T cells [7], facilitates the formation of phagosome in macrophages [10], and participates in the maturation and antigen presentation process of dendritic cells [36].

ZO-2, like MPP1, is a MAGUK protein. ZO-2 facilitates the formation of TJs (Tight Junctions), a sophisticated type of cell junction that maintains cell polarization and confers a selective permeability to certain ions and small molecules in epithelial tissues [37]. Identification of ZO-2 in the interactome of the SARS-CoV-2 E protein correlates with studies that identify the second PDZ of the protein ZO-1 as an interactor of this viral protein using PDZ libraries [38,39]. While ZO-2 functions are well characterized in epithelial cells, no function has been described for this protein in cells of the innate immune system. 

Whereas the previous PDZ proteins exclusively operated as organizers of large scaffolds, SIPA1L1, also known as SPAR1, is a GTPase-activating protein that modulates the activity of the Rap family of small GTPases. With no characterized function in innate immune cells, SIPA1L1 was originally identified in neurons, where it promoted dendritic spine growth as a member of the PSD-95 complex [40]. Additional PDZ interactors of SIPA1L1 include the ephA4 receptor and members of the neurabin family. SIPA1L1 has also been characterized as a target of the viral protein HPV E6 (Table 1).

Another PDZ protein with known functions in the brain, but not described in immune cells, is the protein delphilin (also known as glutamate receptor, ionotropic, delta 2 interacting protein; GRID2IP). This protein belongs to the formin family of cytoskeleton-organizing proteins. Delphilin function has been described in neurons and dendritic spines, in which it binds to receptors like the glutamate receptor GluRd2 [41]. 

Different from the previous PDZ proteins, which have been mainly characterized as organizers of epithelial cell–cell contacts and neuronal synapses, IL-16 is a cytokine with recognized functions in the immune system. Synthesized as a large precursor of around 80 kDa with 4 PDZ domains, pro-IL-16 is cleaved by caspase-3 and secreted as a small peptide of 14 kDa that contains a single PDZ domain. The secreted and mature IL-16 has chemoattractant and growth factor capabilities for CD4^+^ T-lymphocytes, although it also promotes the migration of eosinophils and dendritic cells through direct interaction with their CD4 receptor [42]. IL-16 significantly contributes to pathologies associated with inflammation, like inflammatory bowel disease [43]. Early studies demonstrate IL-16 functions in inhibition of the replication of the human immunodeficiency virus, and in enhancing the host susceptibility to influenza A virus infection [44,45,46].

Our findings indicate that the SARS-CoV-2 E protein can bind to different proteins and that it shares with the SARS-CoV-1 E protein at least one PDZ partner. The sharing of interactors is expected from the high homology between both E proteins; nonetheless, some degree of specificity for partners is also expected. The differences in sequence preceding the PBM in each E protein, together with a cell-specific expression of the potential partners, probably limit the interactions and confer specificity. These specificities might dictate the differences in the evolution of the diseases caused by each virus. Of note is that syntenin was identified as a SARS-CoV-1 E protein binder using two hybrid-based approaches, but has not been identified in subsequent screening studies using libraries that contain individual PDZ domains [38]. This suggests that the use of an in vivo cell strategy resembles better the complexity of the PDZ-PBM interactions.

### 2.5. Expression along Cell Differentiation of the PDZ Proteins Partners of the SARS-CoV-2 E Protein 

We confirmed the endogenous expression of all the identified PDZ protein interactors in THP-1 cells by Western blot analysis. To gain some information about their regulation, we analyzed changes in their expression when monocytes (Mon) were differentiated into macrophage (Mo) or dendritic cells (DC) (Figure 4). All the identified PDZ proteins were expressed in THP-1 Mo, with higher SIPA1L1, ZO-2 and syntenin expression in these differentiation states compared to that observed in Mon (Figure 4A). In contrast, Scrib and Lin-7A expression decreased upon differentiation into Mo. When differentiated into DC, THP-1 cells showed augmented syntenin and delphilin expression compared to Mon. The IL-16 antibody (Ab) recognized mainly the high molecular weight, non-processed form; however, the Ab also recognized some bands of lower molecular weight that probably correspond to the IL-16-processed variants. The low molecular weight bands were more apparent in DC cells (Figure 4A). Scrib was previously described in Mo and DC [47]. We confirmed the expression of additional identified proteins in human primary Mon that were also differentiated into Mo or DC (Figure 4B,C). In general, the tendency in the variation of expression of ZO-2, syntenin and Lin-7A was similar to that observed in differentiated THP-1 cells. ZO-2 and syntenin were expressed in primary Mon, and their expression also increased in Mo and DC. However, the changes in the expression of ZO-2 were more notorious in primary cells than in THP-1 cells. Lin7-A expression was minor in Mo compared to Mon, as observed in THP-1 cells. The increase in the levels of syntenin and ZO-2 in differentiated cells suggests that these PDZ proteins might have a function linked to the specialization of Mo and DC. Different from that observed in the THP-1 model, the IL-16 Ab recognized several bands in primary Mon, suggesting more processing of this protein in primary cells (Figure 4B,C).

We next analyzed the subcellular distribution of syntenin, ZO-2 and IL-16 proteins in THP-1 Mon and derived Mo and DC (Figure 5). In Mon, syntenin staining was intense in small patches, some of which were inside the nucleus, and discrete staining was observed near the plasma membrane (Figure 5A, left, see arrows). ZO-2 also displayed a patched pattern, with marked accumulations in the nuclei (Figure 5B, left, arrows) a localization that agrees with the reported function of ZO-2 in RNA processing in epithelial cells [48,49]. The observed patched patterns of these two proteins coincide with the reported formation of big clusters and/or punctuated patterns that are distinctive of PDZ proteins. IL-16 signal was observed in the cytosol, with a uniform and small dotted pattern (Figure 5C, left, arrows).

PDZ protein’s subcellular distribution was notably different in THP-1-derived Mo and DC. Syntenin staining was distributed near the plasma membrane in Mo (Figure 5A, middle, see arrows), while ZO-2 concentrated in a perinuclear region (Figure 5B, middle, arrows), and IL-16 located in the nuclei of DC (Figure 5C, right, arrows). These variations in the pattern of staining may correlate with changes in the function of these proteins along cell differentiation. No signal was observed when cells were stained without primary Ab as a control (Figure 5D).

### 2.6. The GFP-SARS-CoV-2 E Protein Interacts with the PDZ Proteins Syntenin, ZO-2 and IL-16 

The viral targeting of PDZ domain-containing proteins of the cell junctions might result in their aberrant localization and/or degradation. The analysis by Western blot of cells expressing the SARS-CoV-2 E protein did not reveal significant differences in protein abundance for none of the identified proteins. Studies with SARS-CoV-1 E protein have shown the relocation of PALS1 from the cell junction to the endoplasmic reticulum-Golgi intermediate compartment, which disrupts the polarity complex [31]. We next investigated if the expression of the GFP-SARS-CoV-2 E protein in THP-1 cells alters the previously determined subcellular localization of the endogenous PDZ proteins syntenin, ZO-2 and IL-16.

Expression of the GFP-SARS-CoV-2 E protein induced the relocation of syntenin, and ZO-2 to GFP-positive compartments (Figure 6 and Appendix A). Comparison of the pattern of the PDZ proteins in cells with and without the GFP-SARS-CoV-2 E protein in the same field, (compare cell 1 vs. cell 2), showed changes in the distribution of syntenin and ZO-2 in the presence of E protein. Altered subcellular distribution of the PDZ-protein was particularly evident in the case of syntenin; this protein accumulated at a higher extent to internal regions where the E protein was localized (Figure 6, left), suggesting an E protein-dependent retention of the PDZ protein. ZO-2 nuclear staining decreased, but it accumulated in regions where the E protein was localized (Figure 6, middle). The GFP-SARS-CoV-2 E protein co-localized in small aggregates with IL-16 (Figure 6, right). These data indicated that the GFP-SARS-CoV-2 E protein interacts with endogenous PDZ proteins syntenin, ZO-2 and IL-16, which validates our proteomic findings. 

We then examined the direct interaction of the GFP-SARS-CoV-2 E protein with the 3 selected PDZ partners, as well as the requirement for an intact PBM in the interactions. Either GFP, the GFP-SARS-CoV-2 E protein (wild-type E protein; E wt), or a version lacking the PBM (stop mutant; E stop) were expressed together with tagged versions of syntenin, ZO-2 and IL-16. To avoid the effect of the E protein on cell viability, we first performed the analysis in HEK-293T cells. Pull-down experiments probed the interaction of HA-syntenin with the intact E-protein (E wt), but not with the deletion mutant (E stop), confirming a PDZ-PBM-dependent interaction between these two proteins (Figure 7A, top). Differently, we did not observe specific pull down of the exogenously expressed ZO-2 or IL-16 proteins by the GFP-SARS-CoV-2 E protein, since in these assays both PDZ proteins bind at a similar extent to the GFP, and to both versions of the protein E. Interaction of the GFP-SARS-CoV-2 E protein with HA-syntenin ruled out any problem related to the exposure of the E- protein PBM (as a result of oligomer formation, or diminished accessibility of the C-terminal domain in the context of E-protein insertion in membranes). Moreover, in vitro experiments with isolated PDZ domains have shown a high affinity of a peptide encompassing the PBM of the E protein for the second PDZ domain of ZO-1 (highly similar to that of ZO-2). 

Cell type-specific factors may affect localization and/or potential PDZ-PBM interactions, thus we next expressed the tagged PDZ proteins together with the different GFP-SARS-CoV-2 E protein versions in THP-1 cells. In these assays, we corroborated the GFP-SARS-CoV-2 E protein PBM-dependent binding with syntenin (Figure 7A, bottom), and observed that the GFP-SARS-CoV-2 E protein pulled down ZO-2 and IL-16 (Figure 7B,C). The pulling down of exogenously expressed ZO-2 was PBM specific; however, the interacting pool was small (Figure 7B), and only observed when more astringent conditions for cell lysis were used (RIPA buffer). Regarding IL-16 interaction (Figure 7C), since this cytokine is a protein naturally expressed in THP-1 cells, but not in HEK-293T, it is possible that cell type-specific factors contribute to facilitating the interaction of this cytokine with the SARS-CoV-2 E protein. Nonetheless, in these conditions, the stop mutant of the E protein pulled-down IL-16, although the pulled pool was minor compared to that pulled by the complete version of the E protein. Given this, it is likely that the SARS-CoV-2 E protein interacts with IL-16 in PDZ-dependent and PDZ-independent ways. Additionally, IL-16 (and ZO-2) may bind the SARS-CoV-2 E protein indirectly, through other interactors. 

More studies are necessary in order to characterize the interaction of the SARS-CoV-2 E protein with the PDZ proteins identified in this study, and their possible functional implications in immune cells. Together our data indicate that the SARS-CoV-2 E-protein multifunctionality and/or its diverse effects might be at least partially dependent on its interactions with distinct PDZ proteins in host cells.

## 3. Discussion

SARS-CoV-2 infection can have different outcomes; in most cases, the immune system can resolve infection in a few days, but some patients develop a serious illness that may end in fatal complications. Although not well understood, it is clear that the high morbidity and mortality of COVID-19 disease are linked to dysregulation of the immune response [22]. Monocytes Mo and DC are crucial players in the host defense against viruses and are strong producers of cytokines, which are potential mediators of COVID-19 immunopathology. SARS-CoV-2 infection in these cell types is non-productive; nevertheless, these cells are decisive in promoting virus dissemination and controlling the evolution of COVID-19 disease [50].

The E protein is the smallest of coronavirus’ four structural proteins and, as previously described for the SARS-CoV-1 and the MERS-CoV E proteins, its functions are central in the pathogenesis of the disease [51]. Coronavirus E proteins are integral membrane proteins with two conserved functional features; they can oligomerize forming ion-conductive pores and also interact with PDZ-containing proteins by their PBM. Understanding how these two functions interfere with the responses and cell viability of innate immune cells, and consequently in alterations of the immune response, might provide clues to prevent disease exacerbation and severe complications.

Here, we demonstrate that the expression of the SARS-CoV-2 E protein in THP-1 monocytes caused severe damage in these cells, resulting in most of them dying after 24 h of transfection. Cell death is probably linked to the ion channel function of the E protein. Viroporin E-protein function is a viral strategic feature; viroporins cause host cell swelling and exploding, with the concomitant virion release and dissemination [52]. Assembly of complexes that recruit caspases initiate the degradation of inflammatory cytokines and of the pore-forming protein GSDMD. The pores formed by this protein disrupt the cell membrane, triggering the activation of inflammasomes and the death process coupled with cytokine release [21]. The early degradation of the pore-forming protein GSDMD (observed 6 h after transfection) in the THP-1 cells expressing E protein, is characteristic of inflammasome activation. Our data concur with that of Junqueira et al., who reported that infected monocytes of patients with COVID-19 undergo pyroptosis [19], and support that monocytes play a role in the systemic inflammation observed in COVID-19 pathogenesis. 

Monocyte sensibility to the viral E protein contrasted with that observed in HEK-293T cells, which remained healthy 72 h after transfection. In agreement, Xia et al. reported variations in the sensibility to E-protein expression in some adherent cell lines, including HEK-293 cells, that were resistant [53]. It is very likely that the high sensibility of monocytes to homeostatic disturbances is translated into inflammasome formation. This sensibility is an intrinsic functional characteristic of innate immune cells; monocytes and Mo use inflammasome formation as an effective way of sensing infection and creating a systemic alert to initiate an immune response against pathogens [54]. 

The E protein viroporin function secures virus release, whereas the interactions of its PBM motif warrant an advantageous modification of the host PDZ-interactome. In epithelia, viral disruption of the PDZ interactions controlling polarity favors virus replication and dissemination. In immune cells, viral targeting of PDZ proteins might facilitate the evasion of immune controls [55]. Immune cells rely on the assembly of polarized multiprotein complexes to generate functional outputs like cell activation, differentiation, formation of secretory structures, secretion of cytokines, initiation of phagocytic processes, and migration. The viral targeting of PDZ proteins might be central in the aberrant immune response linked to SARS-CoV-2 infection, including massive and/or imbalanced cytokine secretion, loss of functional capacities of immune cells (like reduced antigen presentation and phagocytosis) and induction of pre-activated states in monocytes. 

Some PDZ proteins recognized by the SARS-CoV-1 and -2 E proteins have been identified using two hybrid assays and screening of libraries of single PDZ domains [15,31,38,39]. Unlike these approaches, in this study, we searched for more physiological conditions by seeking pulled-down endogenous PDZ proteins from human monocytes expressing the intact SARS-CoV-2 E protein. Most of the PDZ proteins identified here are known regulators of polarity in epithelial cells, but never characterized in innate immune cells.

We identified endogenous syntenin as an E-protein interactor and validated this interaction by direct pull-down assays. This direct interaction of the E protein, which bears a classical type II PBM, with syntenin correlates well with the reported ability of this protein to recognize proteins harboring type II PBMs, like syndecans. The functions of syntenin in monocytes are not known, but by analogy with other systems, they are probably related to the regulation of membrane trafficking and exosome secretion. Syntenin in particular stands as an attractive point of intervention in COVID-19 disease. Due to its capacity to regulate the architecture of cell membranes, syntenin is hijacked by different viruses to promote viral transduction, trafficking and dissemination [56,57,58,59,60]. The use of a blocking peptide directed to the first PDZ domain of syntenin inhibits the endosomal entry of different viruses, including that of SARS-CoV-2 [61]. Moreover, syntenin has been characterized as a modulator of TLRs [62]. Then, the E-protein hijacking of syntenin could alter the regulation and function of these receptors.

The findings of our proteomic analysis were validated by the co-localization of IL-16 and ZO-2 with the SARS-CoV-2 E protein and the pull-down experiments that demonstrated a direct interaction between the E protein and these PDZ proteins. Regarding IL-16, this is the first report of a recognition of the viral E protein by this cytokine, which has known functions as a chemoattractant and activator of CD4^+^ T-cells. IL-16 is included in a four-biomarker blood signature that discriminates host systemic inflammation caused by a viral infection from other etiologies [46]. It is noteworthy that a significant increase in the levels of IL-16 is observed in the serum and cerebrospinal fluid of patients with COVID-19 [63]. Increased IL-16-gene expression was suggested as a unique signature linking myocarditis and COVID-19 RNA-mediated vaccination [64]. Concerning ZO-2, we observed that the amount of SARS-CoV-2 E protein and ZO-2 interacting in the co-expression experiments was very small. A possible explanation is that direct interaction is precluded by the native ZO-2 conformation when exogenously expressed in HEK-293T or THP-1 cells. In this regard, experiments with isolated PDZ domains have identified the interaction of the PDZ2 of ZO-1 with the SARS-CoV-2 E protein [38,39]. However, all experiments so far have been performed exclusively with purified PDZ domains, with no data on direct interaction with full-length ZO-1 protein.

Compared to proteins containing class I PBMs, proteins containing class II PBMs display lower affinities for their targets and, as a result, have a minor number of PDZ interactors. Assuming the SARS-CoV-2 E-protein PBM is a class type II, then an interesting possibility is that this protein may recruit, through non-PDZ-based interactions, proteins containing more promiscuous type I PBMs. ZO-2 is a known interactor of Scrib in epithelial cells, and we identified Scrib in our proteomic analysis. Scrib is a PDZ protein with well-recognized functions in epithelia and immune cells; disruption of Scrib complex in epithelial cells affects polarity, while loss of Scrib function in mouse models facilitates the differentiation of M1 inflammatory Mo, but at the same time promotes defects in the defense against bacteria [10]. Scrib targeting in immune cells has been proposed to alter antigen presentation capacity and Mo functionality [55]. Additional studies are needed to explore the endogenous function of the rest of the PDZ proteins identified in this study. An exciting possibility is that, similarly to Scrib, these PDZ proteins orchestrate polarization-dependent functions, such as antigen presentation or phagocytosis, in the context of differentiation processes. A role for some of the identified PDZ proteins in differentiation is supported by variations in their expression and/or subcellular localization along monocyte differentiation to other immune subsets. Syntenin expression for instance augments upon monocytes differentiation into Mo and DC, whereas Lin-7A is highly expressed in monocytes and not in Mo. ZO-2, which also increases in Mo and DC, displays changes in cell distribution, concentrated in nuclei in monocytes, but cytosolic in Mo. The function of these PDZ proteins in cell differentiation is likely since transcriptional profile analysis in Mo and DC identifies variations in the expression of ZO-2, Lin-7A, MPP1 and SIPA1 during the activation of TLRs or phagocytic stimulus [65]. 

The effects of the SARS-CoV-2 E protein over these PDZ proteins deserve further studies to depict possible interrelations with the distinct E protein functional features. In addition to promoting SARS-CoV-2 pathogenesis through its viroporin and PDZ targeting features, the E protein can be recognized by the TLR2, which elicits TLR2-mediated responses [18]. Since this receptor is a trigger of inflammatory responses, the E protein emerges as a potent viral instrument, capable of eliciting inflammation though multiple mechanisms.

## 4. Materials and Methods

### 4.1. GFP-SARS-CoV-2 E Constructs

To build a construct that allows expression of the E protein fused to a GFP-tag at the amino terminal, we used the ORF of the SARS-CoV-2 E protein [22] cloned into vector pDONR207 [23]; Addgene #141273) and recombined it with pEZYeGFP vector ([24]; Addgene #18671) through a LR reaction (recombination between attL and attR sites) using the Gateway system. The GFP-SARS-CoV-2-E construct was verified by enzymatic digestions and sequencing with the vector-specific primer EGFP-C (5′CATGGTCCTGCTGGAGTTCGTG3′). Plasmids were purified using an endo-free maxi-prep purification kit (QIAGEN, Germany). Selected clones were transfected into the indicated cell lines to validate the expression of the recombinant protein.

The GFP-SARS-CoV-2-E plasmid was used as a template to generate a construct lacking the PBM, the stop mutant, or E stop. Mutagenesis reactions using the QuikChange II site-directed mutagenesis kit (Agilent Technologies, CA, USA) were performed according to the manufacturer’s protocol. An early STOP codon to avoid the translation of the PBM was introduced using the specific primers: 

E-STOP forward (5′CAGCTCCAGGGTGCCTTAGCTGCTGGTGTAATACC3′), and E-STOP reverse (5′GGTATTACACCAGCAGCTAAGGCACCCTGGAGCTG3′). The mutant was sequenced to validate the insertion of the mutation.

### 4.2. Cell Culture and Differentiation

HEK-293T cell line was purchased from the ATCC; Caco-2 and THP-1 cell lines were kindly donated by Dr. A. Gonzalez and Dr. H. Reyburn (CNB/CSIC), respectively. 

HEK-293T cell line was maintained in Dulbecco´s modified Eagle medium (DMEM) (Lonza) supplemented with 10% heat-inactivated fetal bovine serum (FBS; Gibco) and 2 mM L-glutamine (Gibco). Caco-2 cells were maintained in the same medium but with 1 mM sodium pyruvate (Gibco). THP-1 cell line was maintained in RPMI 1640 medium (Biowest) supplemented with 10% FBS (Gibco) and 2 mM L-Gln (Gibco). All cell lines were maintained at 37 °C and 5% CO_2_.

THP-1 cells were used as monocytes, or either differentiated to macrophages (Mo) M0, M1 or M2, or dendritic cells (DC), either immature or mature (iDC or mDC), following standard protocols [66,67]. To induce Mo differentiation, 10^6^ cells in exponential growth (0.4–0.5 × 10^6^ cells/mL) were seeded in dishes and cultured in the presence of 100 nM PMA (Phorbol 12-Myristate 13-Acetate; Sigma-Aldrich) for 2 days. Cells were then washed and rested for 24 h to obtain Mo M0, which were either collected or further differentiated as follows. For M1, cells were maintained in media containing 20 ng/mL of interferon γ (IFNγ; PreproTech) and 1 μg/mL of E. coli Lipopolysaccharides (LPS; Sigma-Aldrich) for 6 h; for M2, cells were cultured for 24 h in media with 20 ng/mL of Interleukin 4 (IL-4; PreproTech). For primary cell culture, peripheral blood mononuclear cells (PBMC) were obtained by LymphoprepTM (Alere Technologies, OSL, Norway) density gradient from PBS 1:2 diluted buffy coats of healthy human donors (Community of Madrid (CAM) Transfusion Center). 

Primary myeloid cells were purified from buffy coats obtained from the CAM Transfusion Center. No personal data were registered, and all procedures performed with these cells were under the ethical standards of the Spanish National Centre for Biotechnology (CNB)/CSIC Ethics Committee according to protocols approved for the Ethics Committee of CSIC and the CAM Transfusion Center. CD14^+^ cells were purified using human anti-CD14-labeled magnetic beads and LS columns (Milteny Biotec). Eluted cells were plated (1 × 10^6^ cells/mL) in RPMI 1640 medium supplemented with 10% fetal calf serum, 100 U/mL penicillin, 100 µg/mL streptomycin, and 200 mM L-Glutamine. Generation of Mo and DC was performed using standard protocols. For Mo, CD14+ cells were cultured in the presence of human granulocyte and monocyte colony-stimulating factor (GM-CSF; 10 ng/mL) for 7 days, with additional cytokine supply on days 3 and 6. Mo were challenged with 100 ng/mL of LPS for 24 h. To generate DC, the medium was supplemented with 50 ng/mL of GM-CSF and 25 ng/mL of human IL-4 at days 0 and 3. At day 6, iDCs were harvested and stimulated with 100 ng/mL of LPS for 24 h.

### 4.3. GFP-SARS-CoV-2 E Protein Expression

The construct encoding the viral protein was transiently transfected in HEK-293T cells using LipoD293 reagent (SignaGen Laboratories), and Caco-2 transfection was performed with Lipofectamine 2000 (Invitrogen), in both cases the fabricant procedures were followed. Then, 24 h after transfection the viral fusion protein expression was analyzed by Western blot or immunofluorescence. In the case of the THP-1 line, cells in logarithmic growth (4–5 × 10^5^ cells/mL) were transfected with 30 μg of endo-free plasmid encoding the SARS-CoV-2 E protein by electroporation using the Gene Pulser II (250 mV, 975 µF; Biorad). Cells were collected at different times (0–24 h) to analyze GFP-SARS-CoV-2-E protein expression. In all cases, the pEZYeGFP vector was used as a negative control.

### 4.4. Western Blot 

Cells were lysed (30 min, 4 °C) with NP40 buffer (10 mM Hepes pH 7.5, 15 mM KCl, 1 mM EGTA, 1 mM EDTA, 1% NP40 and 10% Glycerol) or RIPA buffer (20 mM Tris-HCl pH 7.5, 300 mM NaCl, 2 mM EDTA, 1% Triton X-100, 0.1% SDS, 0.5% sodium deoxycholate and 10% Glycerol), both containing protease and phosphatase inhibitors (20 μM leupeptine, 1.5 μM aprotinin, 1 mM PMSF, 1 mM Na3VO4, 40 mM β-glycerophosphate and 2 mM NaF; all from Sigma-Aldrich). Clarified lysates were quantified with pierce 660 nm protein assay (Thermo Scientific). An equivalent protein amount per sample was analyzed by SDS-PAGE, and proteins were transferred to nitrocellulose membranes (Bio-Rad). Ab used for immunoblot were: rabbit anti -GFP, -ZO-2, -Lin-7A, and -SIPA1L1 (all from Invitrogen) and -IL-16, -syntenin, -MPP1, -GSMD, and -Delphilin (all from Abcam), -GAPDH (Santa-Cruz); mouse anti-tubulin mAb (Sigma), -HA mAb (Covance), and -Scrib (Santa Cruz). Primary Ab recognition was visualized using secondary Ab coupled to fluorescent dyes: anti-rabbit IgG StarBright 700 and anti-mouse IgG StarBright 520 (both from Bio-Rad), and anti-goat IgG IRDye 680 (Li-COR). Blots were analyzed with a ChemiDoc MP Imaging System (Bio-Rad). Densitometric analysis of proteins was performed using ImageJ.

### 4.5. Flow Cytometry

Transfected THP-1 cells were recollected at different times and stained with LIVE/DEAD Fixable Violet Cell Stain (Molecular Probes Life Technologies) for 30 min. Cells were washed twice and fixed with 2% PFA. Cells were examined in a Gallios flow cytometer (Beckman Coulter). Data was analyzed with FlowJo software (V.10.2) to determine the percentage of GFP-positive cells and its correlation with death in triplicate samples.

### 4.6. Immunofluorescence Analysis

THP-1 cells, either in exponential growth or previously transfected with the GFP-SARS-CoV-2 E plasmid, were seeded onto untreated coverslips or treated with poli-L-Lysine (Sigma-Aldrich) in p24 plates. Cells were let to attach to the coverslips by incubating for 20 min at 37 °C. The media was gently replaced, and cells were fixed with 2% PFA (10 min, RT). When indicated and after washing three times with PBS, cells were then permeabilized with TX-100 at 0.1% in PBS for 10 min. Cells were washed at least 3 times with PBS and blocked overnight by incubation with 1% Bovine serum albumin (BSA; Sigma-Aldrich) or cell staining buffer (Biolegend). For Ab staining, coverslips were incubated with primary Ab in 0.5% BSA, 0.05% Triton X-100 PBS (Ab solution; 1 h RT), washed and then incubated with a Cy3 fluorophore-tagged secondary Ab against mouse or rabbit IgG (Jackson Immunoresearch; 1 h RT). Primary Ab used is listed in Section 4.4. Nuclei were stained with Hoesh or DAPI dye for 10 min. After extensive washing with PBS, coverslips were mounted using ProLong Glass Antifade Mountant (Life Technologies). 

HEK-293T cells and Caco-2 cells were seeded onto coverslips, 24 h after cells were transfected with the GFP-SARS-CoV-2 E plasmid. The next day cells were either directly fixed or permeabilized to stain nuclei and giantin (rabbit Ab, Biolegend), as indicated above. 

Images were acquired using a confocal microscope Olympus FV1000 and Zeiss LSM 510 Meta. Images were analyzed using the Fiji software (ImageJ, NIH). For co-localization analysis, the JaCoP plugin was used. Mander´s coefficient was calculated to determine the concurrence of the GFP-SARS-CoV-2-E protein with giantin, or with the indicated PDZ proteins. Pearson´s coefficient and Li´s intensity correlation were used to determine the correlation of intensity.

### 4.7. Proteomic Analysis

For protein–protein studies, THP-1 cells were transfected with GFP or GFP-SARS-CoV-2 E. GFP proteins were immunoprecipitated using GFP-Trap Agarose beads (Chromotek, Germany). Cells were lysed (Buffer: 10 mM Tris Cl pH7.5; 150 mM NaCl, 0.5 mM EDTA, 0.5% Nonidet P40 substitute), and GFP proteins precipitated by incubating 1 mg of cell lysates with 20 μL of slurry for 4 h. After extensive washing with buffer lysis, GFP pellets were boiled 10 min with 2X SDS Laemmli Buffer. A tenth part of the immunoprecipitated was used for validation in Western blot. GFP and associated proteins were analyzed by liquid chromatography coupled to Triple TOF Mass Spectrometry in the Proteomics Unit facility at CNB. Two independent experiments with GFP and GFP-E protein precipitates were analyzed separately. Briefly reduced and alkylated samples were loaded into S-Trap columns and trypsin digested. Peptides were separated by their polarity using reverse-phase liquid chromatography and then fragmented using a mass spectrometer Orbital Exploris 240 (LC-MS/MS). MS and MS/MS data for each sample fraction were processed using Analyst TF 1.7 software. Raw data were translated to mascot general file (mgf) format using the PeakView program v.1.2 and a Uniprot database (March 26, 2014) with human taxonomy restriction (NEWT 9606), containing 39,785 protein-coding genes and their reverse entries in an in-house Mascot Server v.2.5.1 (Matrix Science). Search parameters were as follows: fixed modification of carbamidomethyl cysteine; variable modifications of oxidation of methionines and acetylation of the peptide N-termini; peptide mass tolerance ±25 ppm; fragment mass tolerance ± 0.05 Da; and a maximum of 2 trypsin digestion missed cleavages. The accuracy of ±10 ppm was typically found for MS and MS/MS spectra. Criteria to accept individual spectra were based on the Mascot ion score threshold (0.01) as the standard ion score threshold, and the identification certainty was established using false discovery rate criteria (FDR ≤ 1%) for peptide and protein matches using the Scaffold bioinformatic tool v.4.2.1 (Proteome software). This cutoff value for protein identification corresponded to a Mascot score of protein identification of 1519 proteins, from these 1332 were found in the GFP-E immunoprecipitated and 372 proteins were identified only in that of GFP-E protein, in which further analyses were focused. Proteomic results were compared with 155 human proteins containing PDZ motifs to determine the SARS-CoV2 E protein PDZ-dependent interactome.

### 4.8. GFP-SARS-CoV-2 E Protein-PDZ Protein Interaction Validation

HEK-293T and THP-1 cells were transiently transfected with GFP, or the GFP-tagged E protein constructs (E wt or E stop) constructs together with either HA-syntenin construct, kindly provided by Dr. PJ Coffer and described in [68], or Myc-tagged IL-16 or ZO-2 (Origene Technologies). Cells were collected either 6 h (THP-1) or 24 h (HEK-293T) after transfection and lysed with GFP-Trap buffer (see Section 2.3) to evaluate the interactions of GFP-E protein with syntenin and IL-16, or with RIPA buffer (50 mM Tris Cl pH7.5; 150 mM NaCl, 1 mM EDTA, 1 mM EGTA, 1.2% Triton X-100, 0.5% sodium deoxycholate, 0.1% SDS) for the interaction with ZO-2. Cell lysates, from 500 µg to 1 mg, were incubated (4 h, 4 °C) with GFP-Trap Agarose beads (Chromotek, Germany). Immunoprecipitates were washed five times in lysis buffer and diluted in 2× sample buffer. Proteins complexes were eluted by boiling the samples for 10 min and analyzed by Western blot for validation of the interaction.

## Figures and Tables

**Figure 1 ijms-24-12793-f001:**
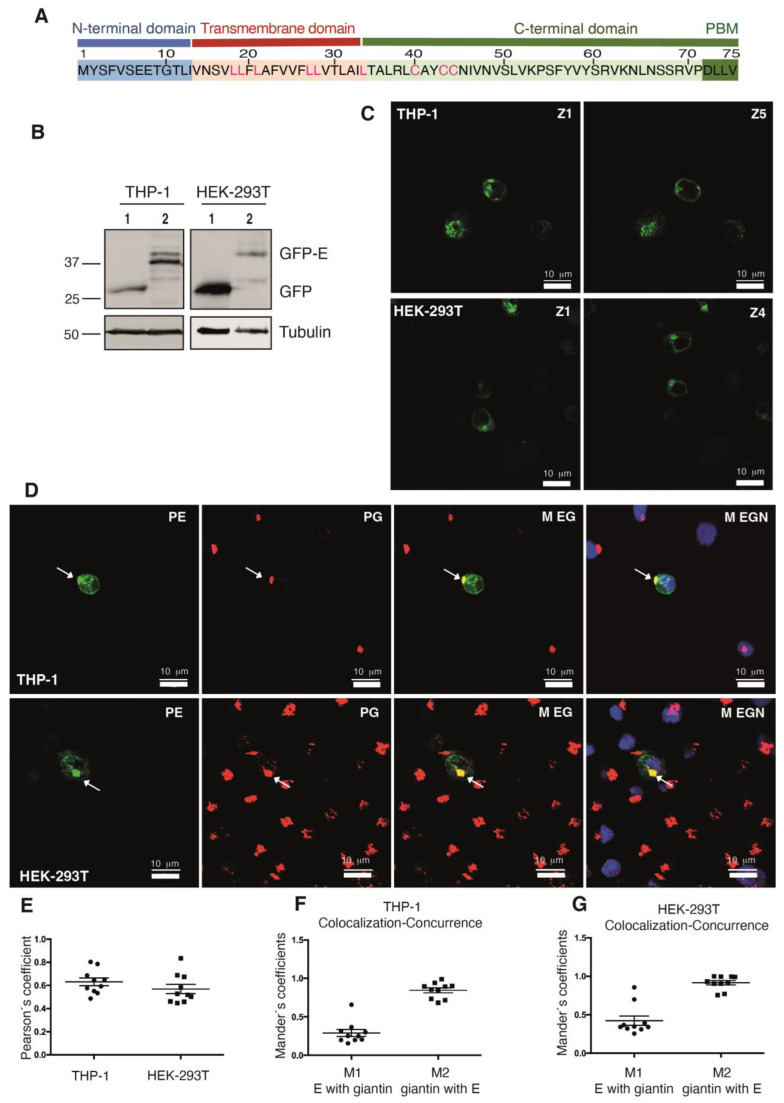
Expression of a recombinant SARS-CoV-2 E protein fused to GFP. (**A**) Aminoacidic sequence of the SARS-CoV-2 E protein. N-terminal, transmembrane and C-terminal domains are indicated. The PDZ binding motif (PBM) is shown in dark green. Palmitoylated-cysteine and leucine residues in the transmembrane region known to modify the migration of the protein in electrophoresis are highlighted in red. (**B**) Cells from the indicated cell line were transfected with either the empty vector (pEZYeGFP, lane 1) or the recombinant plasmid (pEZYeGFP SARS-CoV-2 E, lane 2), lysed and analyzed using a rabbit anti-GFP and a mouse anti-tubulin antibody (Ab). (**C**) THP-1 (**top**) and HEK-293T cells (**bottom**) transfected with the plasmid pEZYeGFP SARS-CoV-2 E were seeded into coverslips and analyzed by confocal microscopy. Different confocal planes (Z) are shown. (**D**) Confocal microscopy images showing the colocalization of the GFP-E protein with the Golgi marker giantin in THP-1 monocytes (**top**) and HEK-293T cells (**bottom**). Cells were transfected with the plasmid pEZYeGFP SARS-CoV-2 E and giantin was detected using a specific Ab and a secondary Ab marked with Cy3 (red), and nuclei were stained with DAPI (blue). P-E, GFP-E projection; P-G, giantin projection; M EG, merge of GFP-E and giantin projections; M-EGN, merge of GFP-E, giantin and DAPI projections. GFP-E is present in different cell regions, including the giantin-positive, Golgi region (arrows). Colocalization of GFP-E and giantin was analyzed in several cells using the JaCoP plugin. (**E**) Pearson’s coefficient (ranging from −1 to 1) was used to determine the fluorescence intensity correlation. (**F**,**G**) Mander´s coefficient (ranging from 0 to 1), using automatic threshold, was used to determine the concurrence of both proteins. M1 corresponds to the fraction of channel A (green; GFP-E) that overlaps with channel B (red; giantin), while M2 corresponds to the fraction of channel B (red, giantin) that overlaps with channel A (green; GFP-E). Analysis shows that E protein partially colocalizes with giantin THP-1 and HEK-293T cells.

**Figure 2 ijms-24-12793-f002:**
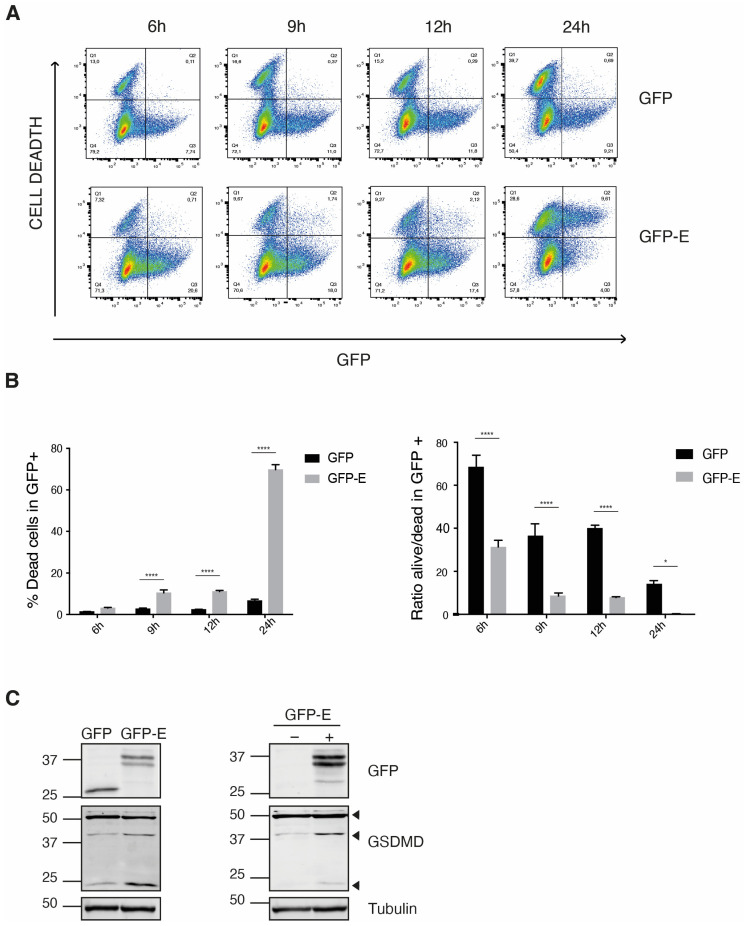
Expression of the GFP-SARS-CoV-2 E protein diminishes the cell viability of THP-1 monocytes. THP-1 cells were transfected with either the pEZYeGFP or the pEZYeGFPSARS-CoV-2 E plasmid and recovered in a culture medium. At the indicated times, cells were harvested, stained with LIVE/DEAD Fixable Violet Cell Stain and analyzed by flow cytometry. (**A**) Bi-parametric analysis of GFP vs. cell death. Histograms of a set of transfections of a representative experiment are shown. The percentage of cells expressing the different combinations of the markers is indicated in each quadrant. (**B**) Analysis of the percentages of dead cells (**left**) and of the ratios of alive to dead cells (**right**) in the GFP-positive cell subsets of a representative experiment. n = 3 independent transfections. Data were analyzed using two-way ANOVA and Bonferroni correction; ns *p* > 0.5, * *p* < 0.05, **** *p* < 0.0001. (**C**) THP-1 monocytes transfected as in A. Cells were lysed 6 h after transfection, and the degradation of Gasdermin D (GSDMD) was determined by Western blot (**left**). After 3 h of transfection with the pEZYeGFPSARS-CoV-2 E plasmid, cells were sorted to separate the GFP negative and GFP positive pools, and then the cells were further recovered for 3 h more and analyzed by Western blot (**right**). Arrows indicated the different degradation fragments of GSDMD. GFP and tubulin were used as expression and loading control, respectively.

**Figure 3 ijms-24-12793-f003:**
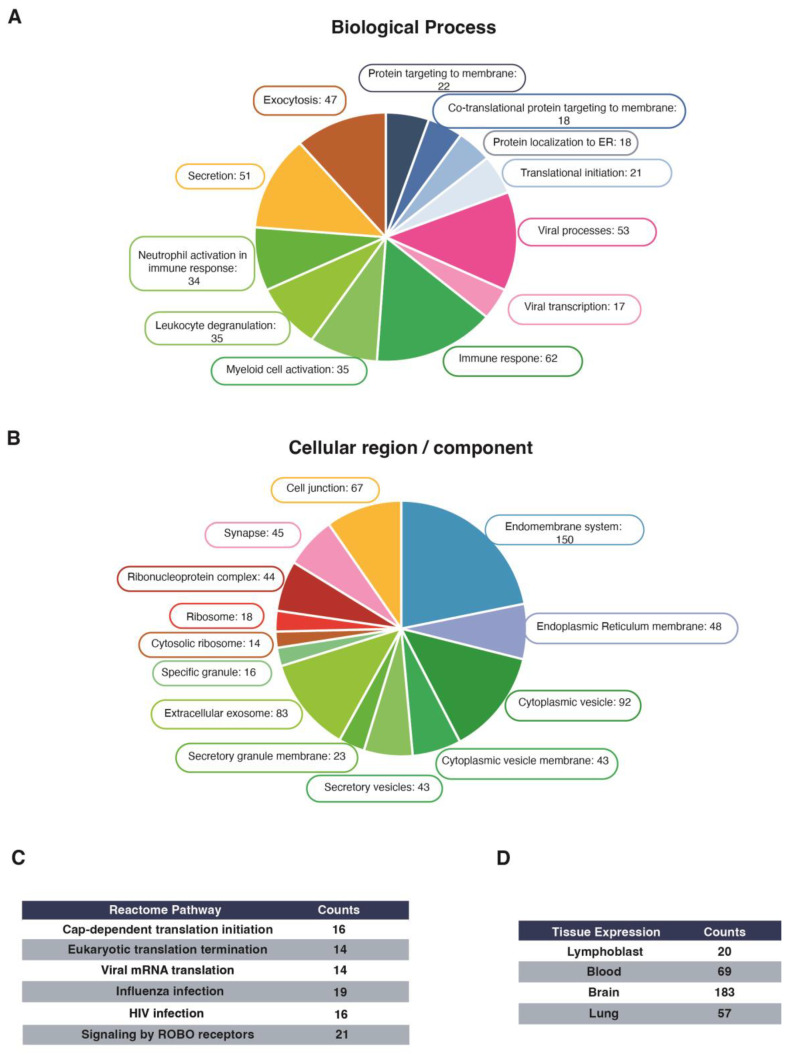
Proteomic analysis of the GFP-SARS-CoV-2 E protein interactors into THP-1 cells. The GFP-tagged E protein was immunoprecipitated using the GFP-trap system. Associated proteins were analyzed by liquid chromatography coupled to Triple TOF Mass Spectrometry. Analysis of the interactome specific for the GFP-E protein in THP-1 cells provided 372 proteins that fall into different functional groups. The interactome was categorized by (**A**) Biological process or (**B**) Cellular/compartment localization. The number of proteins in each category is indicated. Specific interactome was also categorized by a (**C**) Reactome Pathway and (**D**) Tissue Expression.

**Figure 4 ijms-24-12793-f004:**
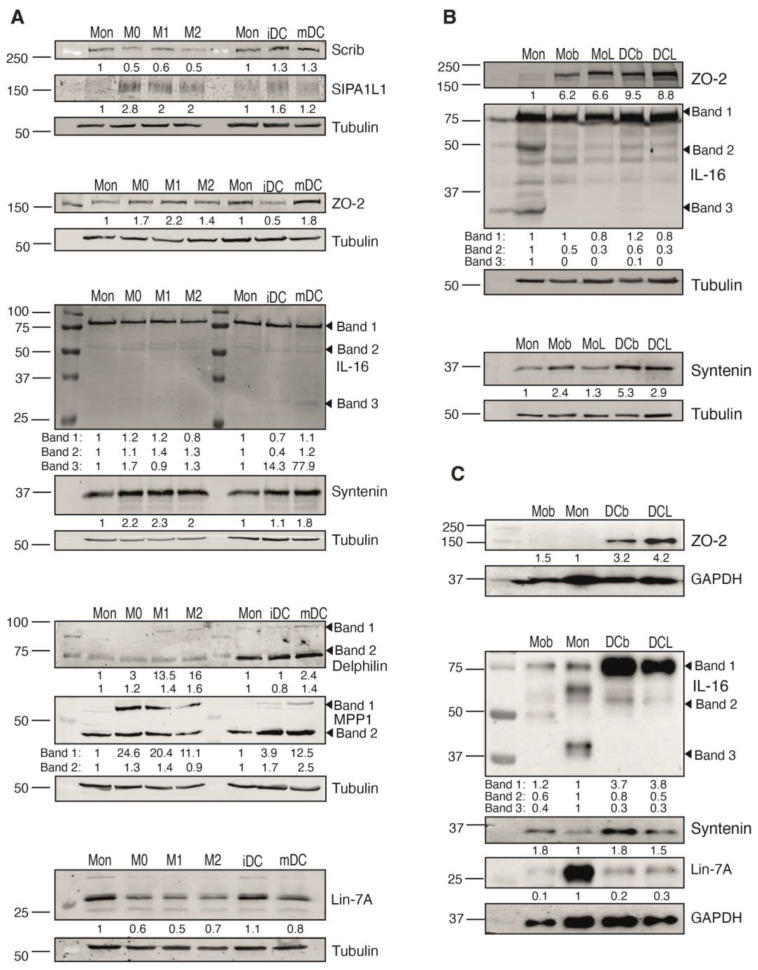
Expression of the PDZ protein interactors of the SARS-CoV-2 E-protein along monocyte differentiation to macrophages or dendritic cells. (**A**) Expression of the PDZ interactors was analyzed by Western blot along the differentiation process of THP-1 monocytic cells. THP-1 monocytes (Mon) were differentiated to either different macrophages (Mo) subtypes (M0, M1 y M2), or to dendritic cells (DC), either immature (iDC) or mature (mDC). (**B**,**C**) Mon from healthy donors were differentiated into either Mo or DC. Expression of the PDZ interactors ZO-2, IL-16, Lin-7A and syntenin was evaluated in the differentiated cells in either basal conditions (Mob and DCb) or after 24 h of a challenge with LPS (100 ng/mL; MoL and DCL). Tubulin (**A**,**B**) or GAPDH (**C**) were used as loading controls. Levels of each protein are indicated beneath the blot. Images shown are representative of different analyses with similar results. Total values were normalized to that of the loading control, and then relativized to the expression of the protein in Mon, which were considered as 1.

**Figure 5 ijms-24-12793-f005:**
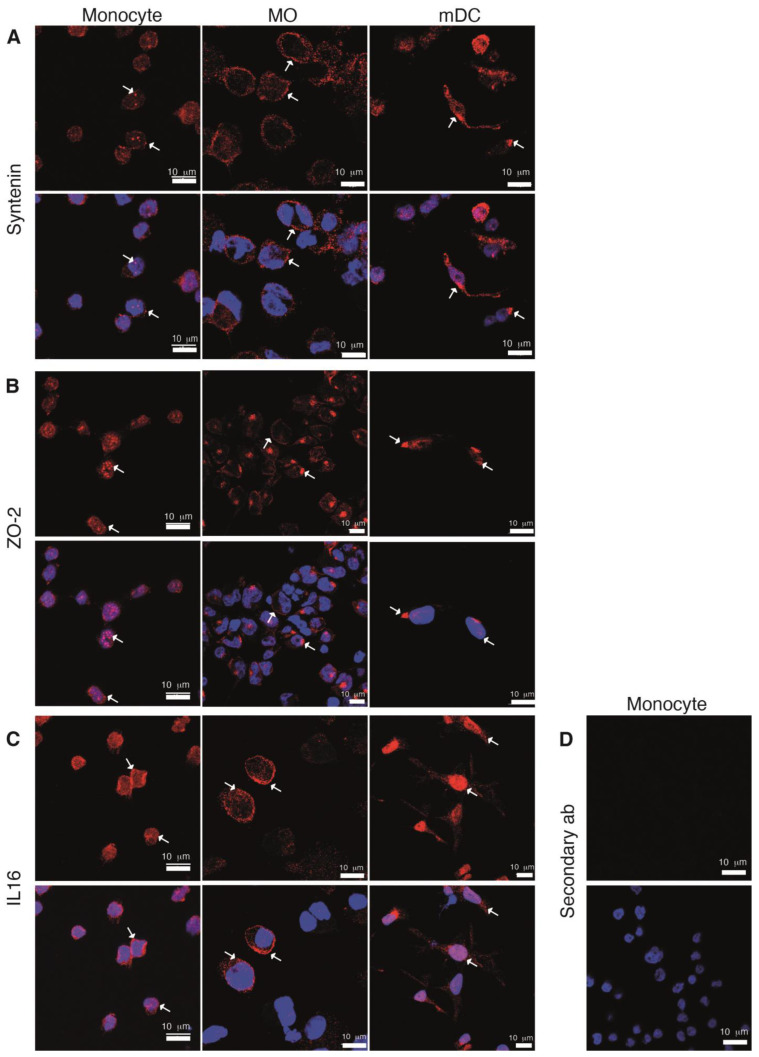
Subcellular distribution of the PDZ protein interactors of the GFP-SARS-CoV-2 E protein in THP-1 monocytes and differentiated cells. The subcellular distribution of syntenin, ZO-2 and IL-16 was analyzed in THP-1 monocytes (left column), or cells differentiated to macrophages (Mo, middle), or dendritic cells (DC, right), by confocal microscopy. A specific primary Ab against either -syntenin (**A**), -ZO-2 (**B**), or -IL-16 (**C**), followed by a secondary Ab marked with Cy3 (red) was used. Nuclei were stained with DAPI (blue). Projection of the different confocal planes for the indicated PDZ protein (**top**), or combined with nuclear staining (**bottom**), are shown. In (**D**), stainings of monocytes with the secondary Ab and DAPI are shown. Images are representative of different analyses with similar results. Arrows indicate particular localizations described in the Section 2.

**Figure 6 ijms-24-12793-f006:**
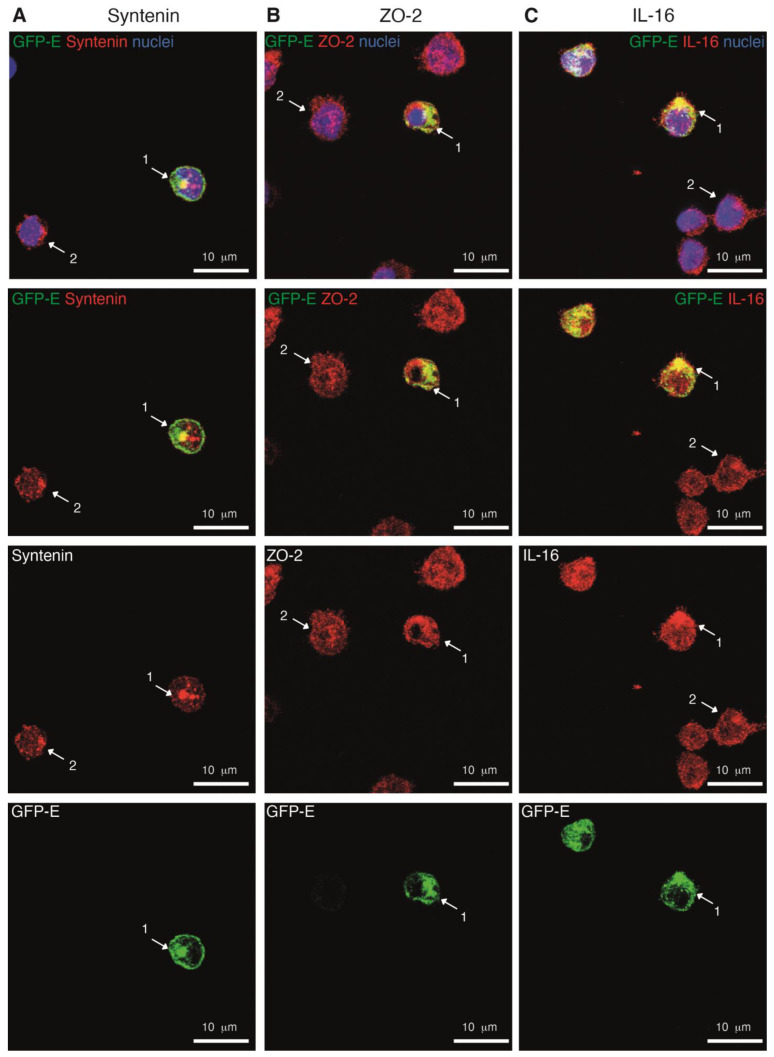
The GFP-SARS-CoV-2 E protein co-localizes with the PDZ proteins syntenin, ZO-2 and IL-16 in THP-1 monocytes. THP-1 cells were transfected with the plasmid pEZYeGFP SARS-CoV-2 E and after 6 h seeded into coverslips. Cells were fixed and permeabilized; specific Ab against the indicated PDZ protein and a secondary Ab marked with Cy3 (red) were used; nuclei were stained with DAPI (blue). Projection of the different confocal planes are shown for E protein (green), for the corresponding PDZ protein (red) and for both proteins and with nuclear staining (top). Intracellular distribution of the GFP-SARS-CoV-2 protein and -syntenin (**A**), left column)), -ZO-2 (**B**), middle column)) and -IL-16 (**C**), right column)) are shown. Images shown are representative of different analyses with similar results. To facilitate comparisons in the distribution of the PDZ protein, arrows indicate the particular localization of the PDZ protein in a cell expressing the E protein (cell 1) and in a cell without the E protein (cell 2).

**Figure 7 ijms-24-12793-f007:**
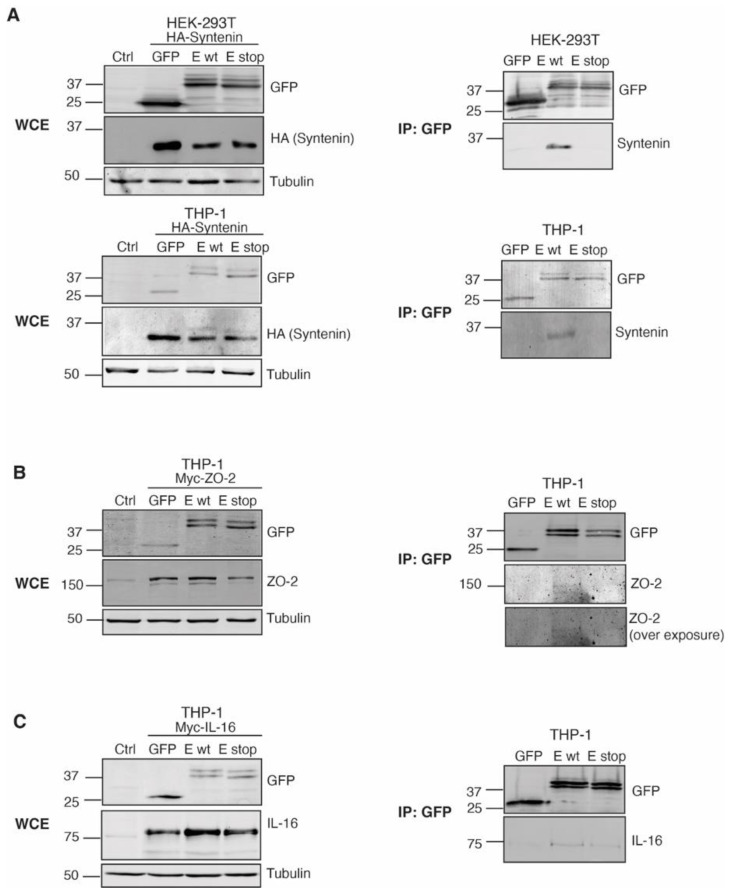
PDZ proteins interact with GFP-SARS-CoV-2 E protein via PDZ/PBM. GFP, GFP-SARS-CoV-2 E protein (E wt) or a version lacking the PBM (E stop) were transfected into the indicated cell line (HEK-293T or THP-1), together with a construct encoding a tagged version of the PDZ proteins (**A**) syntenin (HA-syntenin; (**B**) ZO-2 (Myc-ZO-2); or (**C**) IL-16 (Myc-IL-16.) Cells were lysed (see Section 4) and the whole cell lysate (WCL) was analyzed for expression of the GFP-E versions and for that of the PDZ proteins using the indicated Ab (GFP, HA, ZO-2, or IL-16) by Western blot. Tubulin was used as loading control (left). GFP and GFP-E proteins were immunoprecipitated using the GFP-trap system, and immunoprecipitates analyzed by Western blot (right). In the case of Ab against GFP, syntenin, ZO-2 and IL-16 were used.

**Table 1 ijms-24-12793-t001:** PDZ-containing protein interactors of the GFP-SARS-CoV-2 E protein in THP-1 cells.

	Protein	Function	PDZ	Additional Domains	Viral Interactors
**1**	syntenin-1	Adaptor protein in cell trafficking, exosome biogenesis and tumorigenesis.	2	No.	SARS-CoV-1; VAC FII protein.
**2**	scribble human homolog; Scrib	Scaffold and cell polarity protein in epithelial and neuronal morphogenesis.	4	16 LRR; 3CC.	HPV E6; HTLV-1 Tax; TBEV NS5; ESEV NS1.
**3**	zonula occludens protein 2; ZO-2; tight junction protein 2; TJP-2	Role in tight and adherens junctions.	3	1 SH3; 1 guanylate kinase-like; scribble interacting region.	Adenovirus E4; TBEV NSE5; WNVNS5.
**4**	membrane palmitoylated protein 1; MPP1; 55 kDa erythrocyte membrane protein	Essential regulator of neutrophil polarity.	1	1 SH3, 1 guanylate kinase-like; MPP5 interaction region.	-
**5**	protein lin-7 homolog A; Lin-7A	Establishment and maintenance of the asymmetric distribution of channels and receptors in polarized cells.	1	1 L27; 1 kinase interacting site.	-
**6**	pro-interleukin 16; pro-IL-16	Migratory response in lymphocytes, monocytes and eosinophilsPriming of CD4^+^ T-cells for responsiveness to cytokines. Ligand for CD4. Transcription.	4	Interaction regions with GRIN2A, HTLV-1 Tax; PPP1R12A; PPP1R12B; PPP1R12C.	HTLV-1 Tax; TBEV NS5; WNV NS6.
**7**	signal-induced proliferation-associated 1-like protein 1; SIPA1-like protein 1; SIPA1L1	Reorganization of actin cytoskeleton. Stimulation of GTPase activity.	1	1 RapGAP; 1 potential CC region.	HPV E6
**8**	delphilin; glutamate receptor, ionotropic, delta 2 interacting protein (GRID2IP)	Post-synaptic scaffolding protein. Linking of GRID2 with actin cytoskeleton.	2	1 FH2.	-

## Data Availability

The proteomic datasets for this study can be found in the Proteome Xchange Consortium via the PRIDE partner repository with the dataset identifiers PXD042620 and 10.6019/PXD042620.

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
