# Peer review of "Identification of Host PDZ-Based Interactions with the SARS-CoV-2 E Protein in Human Monocytes"

_ijms, 2023, doi:10.3390/ijms241612793_

Round 1
Reviewer 1 Report
In this manuscript, the authors expressed GFP-fused SARS-CoV-2 E protein in THP-1 monocytes and observed the expression of GFP-fused E protein has affected the viability of the THP-1 cell. Using proteomic analysis, the authors identified eight potential GFP-E PDZ partners. The expression of these 8 endogenous proteins have been confirmed in the THP-1 differentiated cells. The authors also observed the THP-1 cells expressing GFP-E protein interacts with endogenous PDZ proteins and cause cell migrations.
Here are the suggested improvements:
1. The microscopy images are difficult to read, please include additional labels or arrows in the figure and refer the labels on the Figure description section.
2. The proteomic data were never fully validated. The authors used microscopical imaging to show the GFP-E protein expressed in THP-1 cell caused protein relocation, but it will be more straight forward to do westerns to show these eight protein expression were altered due to expression of GFP-E protein in THP-1 cells.
3. Please quantify the western results in Figure 4 to show a more quantified evaluation of the proteins' expression level.
Reviewer 2 Report
Comments:
1. In line of the title- a direct binding experiment should be shown, where the recombinantly expressed SARS-CoV-2 E protein directly interacts to one or more PDZ domain containing proteins from human monocytes. Example- an affinity pulldown when both protein (E protein and the PDZ containing protein) are expressed recombinantly and further purified.
Without this sort of direct experiment, the work seems weak. The western blots as shown in the supplemental files are not always specific- hence a direct protein- protein affinity tag pull down is important.
2. Corresponding to Line 100-101, a figure showing the 75 residues of E protein and the sites of modification should be presented.
3. Line 107-110 does not clearly explain the figure 1B -1C. The protein localization is different between HEK293T and Caco-2 and THP-1. It would be good to have golgi marker controls for caco-2, HEK293T to show that the experimental indeed indicates the localization on golgi.
Round 2
Reviewer 2 Report
-